# Influence of Age on the Standardized Ileal Amino Acid Digestibility of Corn and Barley in Broilers

**DOI:** 10.3390/ani11123575

**Published:** 2021-12-16

**Authors:** Mukti Barua, Mohammad Reza Abdollahi, Faegheh Zaefarian, Timothy J. Wester, Channarayapatna Krishnegowda Girish, Peter V. Chrystal, Velmurugu Ravindran

**Affiliations:** 1Monogastric Research Center, School of Agriculture and Environment, Massey University, Palmerston North 4442, New Zealand; M.Abdollahi@massey.ac.nz (M.R.A.); F.Zaefarian@massey.ac.nz (F.Z.); T.J.Wester@massey.ac.nz (T.J.W.); V.Ravindran@massey.ac.nz (V.R.); 2Department of Animal Science and Nutrition, Faculty of Veterinary Medicine, Chattogram Veterinary and Animal Sciences University, Khulshi, Chattogram 4225, Bangladesh; 3Nutrition and Care, Animal Nutrition, Evonik (SEA) Pte. Ltd., Singapore 609927, Singapore; Girish.channarayapatna@evonik.com; 4Complete Feed Solutions, Howick, Auckland 2145, New Zealand; Peter@completefeeds.co.nz

**Keywords:** age, amino acid, barley, broilers, corn, digestibility

## Abstract

**Simple Summary:**

Efficient amino acid (AA) utilization in broilers is crucial concerning the accuracy of feed formulation, economy of diet and minimizing nitrogenous pollution in the environment. A range of factors (ingredient type, age, sex, feed form and bird type) can affect the AA digestion in poultry. The first week of broiler life is considered the most critical period while chicks source nutrition from residual yolk. Dynamic changes are noted in the digestive tract development, secretion and activity of protein digestive enzymes during the first few weeks in broiler. Limited data exist on the age effect on AA digestibility in broilers, and the results are paradoxical. The aim of the present study was to investigate the influence of age on the standardized ileal digestibility coefficients (SIDCs) of AAs in corn and barley from hatching to the end of the broiler growth cycle (day 42). Based on the results, the age influence on AA digestibility is grain- and AA-dependent. The pattern of the age effect on the SIDC AA in corn was not consistent. In the case of barley, the SIDC AA increased with advancing age.

**Abstract:**

The aim of this study was to determine the standardized ileal digestibility coefficients (SIDCs) of nitrogen (N) and amino acids (AAs) in corn and barley at six different ages (days 7, 14, 21, 28, 35 and 42) of broilers using the direct method. The apparent AA digestibility coefficients were corrected using age-appropriate basal endogenous AA losses. No age effect (*p* > 0.05) was noted for the SIDC of N in corn. The average SIDC of indispensable AAs (IAAs) and total AAs (TAAs) was influenced in a quadratic manner (*p* < 0.05) with the values being higher at day 7 that decreased at day 14, increased and plateaued between days 21 and 35 and dropped again at day 42. The average SIDC of dispensable AAs (DAAs) was influenced linearly (*p* < 0.05). In barley, the SIDC of N and average IAAs, DAAs and TAAs was affected (quadratic; *p* < 0.001) by age. The digestibility increased from day 7 to 21 and then plateaued up to day 42. The present findings confirm that the SIDC of AA in corn and barley are influenced by broiler age and that the age effect on AA digestibility may need to be considered for precise feed formulation.

## 1. Introduction

To improve broiler performance and production economics, accurate feed formulation that closely matches nutrient requirements is crucial. Determination of amino acid (AA) digestibility in feed ingredients is an important way to reach the goal of meeting the AA requirements. Poultry feed formulations based on digestible AA are superior to those based on total AAs because they are reflective of the actual amounts utilized for maintenance and production [1]. A better understanding of the factors influencing AA digestibility is vital to supply available AAs at optimum levels for accurate feed formulation and reduction of diet cost and nitrogen pollution into the environment [2].

The AA digestibility in poultry was measured at the excreta level in the past [3,4] but is currently measured at the terminal ileal level. The excreta digestibility assay has several shortcomings including possible modifying action of hindgut microflora on excreta AA profile and contamination with nitrogen (N) and AAs from urine [5]. Ileal AA digestibility can be categorized as either apparent or standardized/true. For the calculation of standardized ileal digestibility coefficients (SIDCs), the apparent ileal digestibility coefficients (AIDCs) are corrected for basal endogenous AA (EAA) losses originating from various digestive, pancreatic and enzymatic secretions [1]. The SIDC is more additive than AIDC in broiler feed formulations [2].

Grains are the major energy sources in broiler diets. However, they also supply about 40% of the total dietary protein and contribute significantly to the provision of some indispensable AAs (IAAs). Corn (*Zea mays* L.) is used extensively worldwide in poultry diets because of its high palatability, low fiber, high energy and essential fatty acids. Despite the low protein content in corn, owing to its higher inclusion levels (50–70%), it may contribute approximately up to one-third of the protein requirement of broilers. Barley (*Hordeum vulgare* L.) is another grain used in the European Union, Western Canada, New Zealand and Australia. Nevertheless, the inclusion of barley in poultry diets remains limited because of relatively low metabolizable energy, high content of fiber (220 g/kg) and high soluble (45 g/kg) and insoluble (122 g/kg) non-starch polysaccharide (NSP) contents [6]. Because of their viscous nature, the water-soluble fractions of barley exert a negative impact on the digestion and absorption of nutrients, including AAs [7].

Several datasets are available on the AIDC AA in feed ingredients for broilers [8,9]. However, the AA digestibility varies depending on ingredient type [9], class of bird (rooster, broiler, layer) [10] and feed form (mash vs. pellet) [11,12]. Despite the potential effects of age [3,13,14], only sporadic and inconsistent data exist on the age influence on the AA digestibility of ingredients in broilers [3,4,13,15]. Although a number of studies [1,9,12,16] have reported the SIDC AA in a range of feed ingredients, only a few [15,17,18,19,20] exist on the age-related standardized ileal digestibility (SID) of AAs, and the data are limited to two or three specific broiler ages. To the authors’ knowledge, no studies to date have investigated the SIDC AA in grains from hatching to the end of the growth cycle of broilers. The current study was designed to determine the SIDC AA in corn and barley at six different ages (days 7, 14, 21, 28, 35 and 42 posthatch) of broilers.

## 2. Materials and Methods

The experimental procedure complied with the New Zealand Revised Code of Ethical Conduct for the use of live animals for research, testing and teaching and was approved by the Massey University Animal Ethics Committee.

### 2.1. Diets and Experimental Design

Corn and barley were obtained from a commercial supplier and ground in a hammer mill to pass through a screen size of 3.0 mm. Two experimental diets were developed with similar inclusions (938 g/kg) of either corn or barley as the only source of AAs in the diet (Table 1). Titanium dioxide (5 g/kg; Merck KGaA, Darmstadt, Germany) was incorporated in both diets as an indigestible marker. The diets were steam-conditioned at 70 °C for 30 s and pelleted using a pellet mill (Model Orbit 15; Richard Size Limited Engineers, Kingston upon Hull, UK) capable of manufacturing 180 kg of feed/h and equipped with a die ring with 3 mm holes and 35 mm thickness. Pellets were crumbled for the feeding of young chicks during the first two weeks of the experiment.

Representative grain samples were analyzed, in duplicate, for dry matter (DM), N, starch, crude fat, crude fiber, neutral detergent fiber (NDF), gross energy (GE), AAs, calcium (Ca), phosphorus (P) and ash. The AIDC of N and AAs in each grain was determined using the direct method. The AIDCs were then standardized using the age-dependent (days 7, 14, 21, 28, 35 and 42) basal endogenous N and AA losses measured in a previous experiment [21].

### 2.2. Birds and Housing

A total of 696, one-day-old male broilers (Ross 308), obtained from a commercial hatchery, were used in this study. The birds were raised in floor pens and fed a commercial broiler starter diet (12.14 MJ/kg metabolizable energy; 225 g/kg crude protein; CP) from day 1 to 21 and a commercial broiler finisher diet (12.69 MJ/kg metabolizable energy; 190 g/kg CP) from day 22 until day 42 in pelleted form (Table 2).

On day 1, 168 chicks were individually weighed and allotted to 12 cages (14 chicks per cage) in such a way that group mean body weight (BW) per replicate was identical. The remaining chicks were allotted to 12 cages at 5 different ages, namely day 7 (12 birds per cage), day 14 (10 birds per cage), day 21 (8 birds per cage), day 28 (8 birds per cage) and day 35 (6 birds per cage). The test diets were offered for 4 days (days 3–7 and 10–14 (crumbled); days 17–21, 24–28, 31–35 and 38–42 (pelleted)) before collecting ileal digesta on days 7, 14, 21, 28, 35 and 42 posthatch, respectively.

The birds were offered ad libitum feed, and water was freely available throughout the whole experimental period. The room temperature was 32 ± 1 °C in the first week that was gradually reduced to 23 °C by the end of the third week. The floor pens, battery brooders and grower cages were housed in an environmentally controlled room with 20 h of fluorescent illumination per day.

### 2.3. Growth Performance Data

During the 4-day study period, feed intake and BW were recorded on a cage basis each week.

### 2.4. Determination of the Coefficient of Apparent Ileal Digestibility

At the end of each experimental period (days 7, 14, 21, 28, 35 and 42), all birds were euthanized by intravenous injection (0.5 mL per kg BW) of sodium pentobarbitone solution (Provet NZ Pty. Ltd., Auckland, New Zealand). The digesta were collected from the lower half of the ileum and processed as described by Ravindran et al. [8]. The ileum was marked as that portion of the small intestine extending from the Meckel’s diverticulum to a point ~40 mm proximal to the ileocecal junction. In brief, the ileum was excised and divided into halves (proximal and distal ileum), and the digesta samples were collected from the lower half toward the ileocecal junction after gently flushing with distilled water into plastic containers. The ileal digesta from birds within a cage were pooled after collection, frozen immediately and then lyophilized (Model 0610, Cuddon Engineering, Blenheim, New Zealand). Diet and lyophilized digesta samples were ground to pass through a 0.5 mm sieve and stored in airtight plastic containers at 4 °C pending analysis. 

### 2.5. Gizzard pH and Jejunal Digesta Viscosity

From the birds euthanized for ileal digesta collection, two birds from each replicate cage were used for the measurement of gizzard pH by a digital pH meter (pH spear, Oakton Instruments, Vernon Hill, IL, USA). The glass probe was inserted through an opening made in the gizzard and was placed directly in the digesta. Three values were taken from the proximal, middle and distal regions, and the average value was considered as the final pH value. The jejunal digesta viscosity was also determined from these birds. The digesta was collected from the distal jejunum, followed by centrifugation at 3000 × *g* at 20 °C for 15 min. A 0.5 mL aliquot of the supernatant was used in a viscometer (Brookfield digital viscometer, Model DV2TLV; Brookfield Engineering Laboratories Inc., Stoughton, MA, USA) fitted with CP-40 cone spindle with shear rates of 5–500/s to measure the digesta viscosity.

### 2.6. Chemical Analysis

Dry matter was measured using the standard procedure (Method 930.15) [22]. Titanium was analyzed on a UV spectrophotometer (Berthold Technologies GmbH and Co. KG, Bad Wildbad, Germany) following the method described by Short et al. [23]. Gross energy was determined by an adiabatic bomb calorimeter (Gallenkamp autobomb, Weiss Gallenkamp Ltd., Loughborough, UK) standardized with benzoic acid. Starch was analyzed using the Megazyme Total Starch Assay kit (Megazyme International Ireland Ltd., Wicklow, Ireland) based on thermostable α-amylase and amyloglucosidase [22,24]. Nitrogen was determined by combustion (Method 968.06) [22] using a carbon nanosphere-200 carbon, N and sulfur autoanalyzer (LECO Corporation, St. Joseph, MI, USA). The CP content was calculated as N × 6.25. Fat was determined using the Soxhlet extraction procedure (Method 2003.06) [22]. Neutral detergent fiber was determined (Method 2002.04) [22] using Tecator Fibertec^TM^ (FOSS Analytical AB, Höganäs, Sweden). Ash was measured by ashing in a muffle furnace at 550 °C for 16 h (Method 942.05) [22]. Calcium and phosphorus concentrations were measured by Inductively Coupled Plasma-Optical Emission Spectroscopy (ICP-OES) using a Thermo Jarrell Ash IRIS instrument (Thermo Jarrell Ash Corporation, Franklin, MA, USA).

Amino acids were analyzed following standard procedures (Method 994.12) [25]. Briefly, the samples were hydrolyzed with 6 N HCl containing phenol for 24 h at 110 ± 2 °C in glass tubes in an oven. Amino acids were measured using AA analyzer (ion exchange) with ninhydrin post-column derivatization. The chromatograms detected at 570 and 440 nm were integrated using dedicated software (Agilent Open Lab software, Waldbronn, Baden-Württemberg, Germany). Cys and Met were analyzed as cysteic acid and methionine sulphone, respectively, by oxidation with performic acid–phenol for 16 h at 0 °C prior to hydrolysis. For the measurement of Trp, the samples were saponified under alkaline conditions with barium hydroxide solution in the absence of air at 110 °C for 20 h in an autoclave. The internal standard α-methyl Trp was added to the mixture following hydrolysis. After adjusting the hydrolysate to pH 3.0 and diluting with 30% methanol, Trp and the internal standard were separated by reverse phase chromatography (RP-18) on an HPLC column (CORTECS C18 Column; 2.7 µm, Waters Corporation, Dublin, Ireland). Finally, detection was selectively performed by means of a fluorescence detector to prevent interference by other AAs and constituents.

### 2.7. Calculations

Data were expressed on a DM basis. The AIDCs of AAs were calculated from the dietary ratio of AA to Ti relative to the corresponding ratio in the ileal digesta using the following formula.
AIDC of AA = [(AA/Ti)_d_ − (AA/Ti)_i_]/(AA/Ti)_d_
where (AA/Ti)_d_ = ratio of AA to Ti in the diet, and (AA/Ti)_i_ = ratio of AA to Ti in the ileal digesta.

Apparent digestibility values for N and AAs were then standardized using the age-appropriate basal endogenous N and AA estimates (EAA; grams per kilogram of DM intake (DMI)) analyzed at different ages (days 7, 14, 21, 28, 35 and 42) in a previous experiment [21].
SIDC = AIDC + [Basal EAA (g/kg DMI)/Ing. AA (g/kg DM)]
where SIDC = standardized ileal digestibility coefficient of the AA, AIDC = apparent ileal digestibility coefficient of the AA, Basal EAA = basal endogenous AA loss, and Ing. AA = concentration of the AA in the ingredient.

### 2.8. Data Analysis

Cage was considered as the experimental unit. Data were analyzed by the GLM procedure of SAS (version 9.4; 2015; SAS Institute, Cary, NC, USA) for each grain. Differences were considered significant at *p* < 0.05. Orthogonal polynomial contrasts were performed to determine the linear and quadratic effects of age. The relationships between SIDC AA and other parameters were analyzed by Pearson correlation. 

## 3. Results

### 3.1. Proximate and Nutrient Composition

The proximate and nutrient composition of the grains is summarized in Table 3. The results are presented on an “as-received” basis.

In both grains, starch was the main component followed by NDF in corn and CP in barley. The starch content was higher in corn than in barley, and the opposite was observed for the CP content. The NDF in corn was determined to be 83.1 g/kg which was lower than that in barley (110 g/kg). The contents of Ca in both corn (0.17 g/kg) and barley (0.14 g/kg) were negligible.

Among the IAAs, the content of Leu was the highest followed by Val, Arg, Ile and Thr in both grains, whereas lower contents were determined for Trp and Met. The Glu was the major dispensable AA (DAA) followed by Pro in both corn and barley. The variations in CP contents between the two grains were reflected in total AA (TAA) contents with a higher value in barley (92.5 g/kg) compared to corn (61.8 g/kg).

### 3.2. Growth Performance, Gizzard pH and Jejunal Digesta Viscosity

Weekly data on the performance, gizzard pH and jejunal digesta viscosity of birds fed corn- or barley-based diets are presented in Table 4.

Mortality during the experiment was negligible. Out of the 696 birds, only four died, and the deaths were not related to any specific treatment. The daily feed intake (DFI) and daily weight gain (DWG) increased (quadratic; *p* < 0.001) in both corn- and barley-based diets as birds grew older. Gizzard pH increased in a quadratic manner (*p* < 0.001) with advancing age in both grains. A decline in gizzard pH was observed from day 7 to day 14 but increased beyond day 21. The jejunal viscosity in corn was unaffected (*p* > 0.05) by age. In the case of barley, however, the jejunal digesta viscosity was influenced quadratically (*p* < 0.05) by age. Higher viscosity was observed on days 7 and 42 (2.94 cP). After day 7, a reduction in viscosity was observed at day 14 that plateaued until day 35. A further increase was observed at day 42.

### 3.3. Ileal Digestibility Coefficients of N and AAs in Corn

The influence of broiler age on the AIDC, SIDC and SID content of N and AAs in corn is presented in Table 5, Table 6 and Table 7, respectively. 

A quadratic increase (*p* < 0.001) was observed for the AIDC of N and average digestibility of IAAs, DAAs and TAAs of corn with the advancing age of broilers (Table 5). The AIDC of N and average AIDC of IAAs, DAAs and TAAs increased from day 7 to 21 then plateaued up to day 42. The AIDC of all individual IAAs increased in a quadratic manner (*p* < 0.001) with age. With the exception of Cys (*p* > 0.05), an increase (quadratic; *p* < 0.001) was observed for the AIDC of all individual DAAs.

The SIDC of N in corn was unaffected (*p* > 0.05; Table 6) by age. Bird age, however, quadratically influenced the average SIDC of IAAs (*p* < 0.002) and TAAs (*p* < 0.05). The higher values were recorded on day 7 than day 14, and the SIDC values increased until day 21 and plateaued until day 35, followed by a decrease on day 42. The SIDC of average DAAs was influenced in a linear (*p* < 0.05) manner with a higher value on day 7 (0.881) than day 14 (0.788). Afterward, an increase in the SIDC was observed on day 21 which plateaued until day 35, followed by a decrease on day 42. Except for Thr (*p* > 0.05), the SIDC of all individual IAAs and DAAs was influenced (linear or quadratic; *p* < 0.05 to < 0.001) by broiler age. 

No age influence (*p* > 0.05) was recorded for the SID protein content of corn (*p* > 0.05; Table 7). The SID contents of total IAAs, DAAs and total AAs was influenced by age (quadratic; *p* < 0.05 to < 0.001). The SID content of total AAs was higher at day 7 (54.1 g/kg), increased from day 14 to 21 and plateaued until day 42. The SID contents of all individual AAs, except Thr and Ser, were influenced quadratically (*p* < 0.05 to < 0.001) by bird age. A linear pattern (*p* < 0.001) was observed for age effect on the SID content of Ser.

### 3.4. Ileal Digestibility Coefficients of N and AAs in Barley

The impact of broiler age on the AIDC, SIDC and SID content of N and AAs in barley is presented in Table 8, Table 9 and Table 10, respectively. 

The AIDC of N and average digestibility of IAAs, DAAs and TAAs increased (quadratic; *p* < 0.001) from days 7 to 21 and then plateaued from days 21 to 42 (Table 8). The AIDC of all individual AAs in barley increased in a quadratic manner (*p* < 0.001) as the birds grew older.

The SIDC of N and average SIDC of IAAs, DAAs and TAAs increased quadratically (*p* < 0.001) with the advancing age of broilers (Table 9). The values increased from days 7 to 21 and then plateaued until day 42. The lower SIDC of TAAs was recorded at day 7 (0.617), followed by day 14 (0.738). The SIDC of every single AA in barley increased (quadratic; *p* < 0.05 to 0.001) with bird age with lower values on day 7.

The SID protein content of barley increased (quadratic; *p* < 0.001) from 77.7 g/kg on day 7 to 94.1 g/kg on day 21 and then plateaued from days 21 to 42 (Table 10). The SID contents of total IAAs, DAAs and total AAs increased (quadratic; *p* < 0.001) from day 7 to 21 and then plateaued until day 42. The SID contents of all individual AAs increased in a quadratic manner (*p* < 0.05 to 0.001) with lower values on day 7, which increased either at day 14 or 21 and then plateaued until day 42.

### 3.5. Uplift in Digestibility Coefficients Due to Correction for Age-Appropriate Endogenous Amino Acid Losses 

The percentage increase in the digestibility coefficients of N and AAs after standardization of apparent values for basal endogenous N and AA losses is shown in Table 11.

The correction of AIDC for age-appropriate endogenous N and AA losses resulted in an increase in the SIDC regardless of age, though the extent of increase reduced as the birds grew older. After standardization of the AIDC estimates, the average TAA digestibility coefficients increased in corn by 32.5% (day 7), 17% (day 14), 13.9% (day 21), 14.7% (day 28), 14.2% (day 35) and 9.85% (day 42). In the case of barley, the corresponding increases were recorded as 31.6% (day 7), 11.5% (day 14), 10.0% (day 21), 10.8% (day 28), 10.0% (day 35) and 7.17% (day 42).

## 4. Discussion

Most available data on the AIDC and SIDC AA of ingredients have been determined using older broilers (22 to 35 days of age), and the estimates are applied in feed formulations regardless of broiler age. Several reports exist on age-related AA digestibility in ingredients for poultry, but the results are contradictory and inconclusive. Some studies have documented reductions in protein or AA digestibility [3,4] with advancing age, while others reported increases in digestibility estimates [13,26]. The present experiment aimed at identifying whether the broiler age has an impact on the SIDC of AAs in corn and barley.

Although it may be intuitively expected that the AA digestibility in broilers will vary depending on broiler age, studies comparing the SID of AAs corrected using age-appropriate EAAs are limited [18,19,20]. A previous study in our laboratory [14] reported the SIDC AA in wheat and sorghum at six different ages (days 7, 14, 21, 28, 35 and 42) of broilers. In the current research, the AIDC and SIDC AA in corn and barley were determined from hatching to the end of the growth cycle of broilers, and the AIDC values were standardized using age-appropriate basal EAA losses.

### 4.1. Nutrient Composition

The proximate nutrient contents of corn and barley were comparable to the values reported previously [12,16,27]. The higher starch content in corn (590 g/kg) than barley (541 g/kg) was expected. Due to high starch (620 to 720 g/kg) and crude fat (34 to 52 g/kg) contents in corn, it contains higher energy than any other grain. The CP content of corn (67.8 g/kg) was lower than the range (71 to 94 g/kg) reported by Cowieson [28].

The CP content (115 g/kg) in barley was marginally higher than the value (101 g/kg) reported by Perera et al. [27] and lower than the range (121 to 180 g/kg) of Bandegan et al. [16]. The higher CP content in barley compared to corn was in agreement with previous studies [9]. The AA contents of corn and barley were identical or close to those reported in previous studies [9,12,16,27].

### 4.2. Performance, Gizzard pH and Jejunal Digesta Viscosity

As anticipated, regardless of the grain type, an increase in both the DFI and DWG was observed with advancing age. The gizzard pH on day 7 of birds fed corn (2.53) and barley diets (2.14) was close to the values of 2.39 and 2.33 observed by David et al. [29] and Morgan et al. [30], respectively, for broilers of similar age. The reduction of gizzard pH at day 14 compared to day 7 was in line with the findings of David et al. [29] when feeding a corn-based diet. Based on the review of 15 published studies, Angel et al. [31] also reported a reduction in gizzard pH in broilers at day 14. According to Rynsburger [32], the secretion of gastric acid in the proventriculus increased from day 2 to 15, causing a decrease in pH.

As observed by Nitsan et al. [33], the secretion and activity of digestive enzymes and hydrochloric acid (HCl) secretion from proventriculus increase with broiler age. However, the observed increase in gizzard pH with bird age after day 14 in both diets could be, at least in part, explained by the increasing intake of feed with neutral pH with advancing age [34]. An increased feed load can dilute the HCl secreted and consequently increase the gizzard pH. The secretion of HCl is fundamental to sustain an acidic environment and to convert pepsinogen to pepsin, the first step in protein digestion [32]. Thus, the implication is that a high gizzard pH would compromise the protein digestion. In this experiment, however, gizzard pH was not correlated (r = 0.242; *p* > 0.05) with the average SIDC of TAAs in corn. On the contrary, in the case of barley, there was a positive correlation (r = 0.644; *p* < 0.001) between the gizzard pH and the average SIDC of TAAs. Therefore, it could be speculated that, though the gizzard pH was elevated by age in barley, contrary to expectations, AA digestibility also increased. Besides the pH, there are several other factors, to be discussed later, that potentially influence the AA digestibility in broilers.

The jejunal digesta viscosity in corn was not influenced by age, and no correlation (*p* > 0.05) existed between the digesta viscosity and the average SIDC of TAAs. In barley, the range (2.69–2.94) of jejunal digesta viscosity at different ages was notably higher than that of corn (2.03–2.34), with viscosity being higher on days 7 and 42. Yu et al. [35] replaced corn with increasing barley inclusions (0, 125, 250, 500 and 1000 g/kg) in 3-week-old broilers and reported an increase in the duodenal digesta viscosity from 1.46 cP (0 g/kg) to 2.40 (125 g/kg), 2.15 (250 g/kg), 2.71 (500 g/kg) and 2.81 cP (1000 g/kg). A positive relationship exists between the soluble NSP content and digesta viscosity [7]. Corn contains negligible amounts of soluble NSP compared to wheat and barley [28]. The NSP in plant cell walls such as β-glucans, the major NSP in barley, and the pentosans of rye and wheat exhibit significant antinutritive effects in poultry [36]. A portion of NSP of high molecular weight dissolves in the gastrointestinal tract, increasing the viscosity of gut contents that impedes the digestion and absorption of nutrients [37]. In the case of barley, however, the jejunal digesta viscosity tended (r = −0.292; *p* = 0.084) to be negatively correlated with the average SIDC of TAAs.

Besides the soluble NSP content, a myriad of factors such as growing location, storage time, ingredient inclusion level, age of bird, heat processing and pelleting temperature have been shown to influence the digesta viscosity in barley-containing diets [6]. It is difficult to explain the high viscosity at day 42 observed in the current work since previous studies [38] have reported decreased intestinal viscosity in older birds fed barley-based diets. During the evaluation of a high viscosity hull-less barley, Salih et al. [38] recorded a drop in the digesta viscosity in broilers from two weeks (2.59 cP) to eight weeks of age (1.74 cP).

### 4.3. Ileal Digestibility Coefficients of Nitrogen and Amino Acids

The first week is the most critical period in a bird’s life when they consume only small amounts of feed and depend mostly on the nutrients from residual yolk [39]. Notable changes occur in the morphology and development of the gastrointestinal tract during the first few weeks of life that contribute to improved digestion and absorption of nutrients. During the first few weeks, there is high protein demand for the development of organs and muscle [39]. The secretion and activity of different proteolytic enzymes such as trypsin, chymotrypsin, intestinal peptidase and dipeptidase also generally increase with age [33].

From previous studies, it is evident that the AIDC of AAs is variable depending on broiler age [3,13,40,41]. In the current experiment, with advancing broiler age, an increase in the AIDC of N, all individual AAs (except Cys) and average of TAAs was observed for corn. Compared to day 7, the average AIDC of all AAs increased by 0.46, 22.0, 19.4, 18.6 and 20.5% at days 14, 21, 28, 35 and 42, respectively. The increased AIDC AA with age is in agreement with previous findings [13,19,26]. An increase in ileal N digestibility from 78% at day 4 to about 90% at day 21 has been reported in broilers fed diets based on corn–soybean meal by Noy and Sklan [40]. They concluded that the hydrolysis of exogenous and endogenous proteins was not optimum due to insufficient proteolytic activity at the early posthatch period. An increase in apparent AA digestibility from 1 to 10 days of age was reported with a corn–soybean meal diet by Batal and Parsons [41]. Wallis and Balnave [26] recorded an increase in AIDC AA from day 30 to 50 posthatch (0.732 vs. 0.814) feeding a finisher diet containing a wide range of feed ingredients (wheat, sorghum, soybean meal, cottonseed meal, meat and bone meal, poultry tallow, poultry offal meal, feather meal). Huang et al. [13] determined the AIDC AA of eight ingredients (corn, wheat, sorghum, soybean meal, canola meal, meat and bone meal, cottonseed meal and millrun) at three broiler ages (days 14, 28 and 42). Combining all the results, it was concluded that the digestibility increased with advancing age. The trends, however, were variable depending on the AA and ingredient type. In their study, higher AIDC AA was recorded at days 28 and 42 compared to day 14 in the case of corn, soybean meal, canola meal and meat and bone meal. The AIDC in millrun at day 42 was higher than those at 14 and 28 days. Whilst there was no age influence on the AIDC of most AAs in cottonseed meal, Lys and Arg digestibility increased with age. On the contrary, in wheat, the AIDC of most individual AAs was recorded to be higher at day 14 than at days 28 and 42. In sorghum, the AIDC AA was higher at day 42 compared to day 28 but similar to those at day 14 with the exception of His, Lys, Ser and Gly which were higher at day 42.

In the present study, with the exception of Thr, linear or quadratic responses to broiler age were observed for the SIDC of all individual AAs in corn. Unlike the AIDC, the pattern of increase in SIDC AA in corn was not gradual with increasing age. Rather, a decline was observed from day 7 to 14 followed by an increase from day 14 to 21, a plateau between days 21 and 35 and a decline at day 42.

Differing age-related trends between the AIDC and SIDC estimates have also been observed by Adedokun et al. [19]. These researchers, comparing AA digestibility between days 5 and 21 of broilers for five ingredients (corn, light and dark distiller’s dried grains with solubles, canola meal and soybean meal), reported an increased AID AA with age in all test ingredients. However, increasing broiler age elevated the SID AA only in corn and distiller’s dried grains with solubles and had no influence on the SID AA in soybean meal and canola meal.

Apart from the age effect, another notable observation was made on the AA digestibility in corn in the current study. Though maize contained lower CP and TAA contents compared to barley (Table 3), the average SIDC in maize was 39.4% (day 7), 3.4% (day 14), 9.6% (day 21), 10.8% (day 28), 6.9% (day 35) and 8.5% (day 42) higher than barley. Compared to other grains, maize contains low levels of soluble NSP (1 g/kg) and highly digestible nutrients for broilers. Barley contains a high amount of soluble NSP (45 g/kg) compared to maize [28]. According to Andriotis et al. [42], the barley endosperm is composed of β-glucans (70%) and arabinoxylans (20%). The high content of soluble β-glucan is the major antinutritional factor in barley, and, to assess the potential feed value of barley for poultry, determination of the content and properties of β-glucan is crucial [27]. When birds are fed diets with high inclusions of barley, the NSP impedes nutrient digestion and absorption by two mechanisms. First, the soluble NSP forms gel-like viscous matter, which impairs the interaction between nutrient substrates and endogenous enzymes. Second, the insoluble NSP fraction exerts a “cage effect” by encapsulating the nutrients (starch, protein) in endosperm cells, impeding the contact between nutrients and digestive enzymes. High levels of β-glucans result in thicker cell walls than low levels of β-glucans [43,44]. Though barley has a potential to be included in broiler diets without compromising growth performance [45,46], its inclusion is not recommended for young birds (<7 days) in this experiment due to very low SIDC AA at day 7.

In the present study, an increase in the AIDC of N and average AIDC of IAAs, DAAs and TAAs with age was observed in barley. The AIDC of all AAs increased from day 7 to 21 and then plateaued until day 42. Similar patterns of increase were observed in the SIDC of N, SIDC of all individual AAs and average SIDC of IAAs, DAAs and TAAs with lower values on day 7, then an increase until day 21, which plateaued from day 21 to 42. The average SIDC of TAAs in barley at day 7 was 16.4% lower than day 14, 25.0% lower than day 21 and 23.4% lower than the average of days 28, 35 and 42. Szczurek et al. [20] determined the AA digestibility in three grain sources (barley, triticale and wheat) in broilers at day 14 vs. 28 posthatch. In barley and triticale, the average AIDC of IAAs was notably higher at day 28 than 14. The average AIDC of IAAs in barley was reported to be 0.730 on day 14 and 0.780 on day 28. The SIDC of all AAs in barley and most AAs in triticale was also higher at day 28. In the case of wheat, however, no difference was recorded for the average AIDC of IAAs between the two ages. The effect of age on the AA digestibility in wheat and sorghum at six different ages (days 7, 14, 21, 28, 35 and 42) of broilers has already been documented [14]. The AIDC of all individual AAs in wheat increased as the birds grew older. Although the average SIDC of TAAs was unaffected by age, the digestibility of some individual AAs (Met, Trp, Asp and Cys) was higher in the older birds. It is possible that the GIT of older birds is developed sufficiently enough to counteract any adverse effects of digesta viscosity induced by β-glucan. Similar observations were made in other studies [38,47], indicating that the digestibility of nutrients in barley diets improves with age and that viscosity is not a limiting factor in older birds.

Several factors may explain the increased AIDC AA with advancing age for corn and barley. Lower production and activities of pancreatic enzymes are a constraint for nutrient digestion in young chicks [39]. Nitsan et al. [33] measured the activities (units/kg BW) of trypsin and chymotrypsin in the pancreas and small intestine from hatching to 23 days of age. The activity of trypsin was low at days 3 to 6 posthatch and increased on day 14. A gradual increase in chymotrypsin activity was also observed from hatch to day 14, and this remained constant afterward. Maximum activities of trypsin and chymotrypsin were observed on day 11. In the intestinal contents, the activity of trypsin increased 10-fold from hatching to day 14, and that of chymotrypsin increased 3-fold by day 20. As stated by Tarvid [48], the total intestinal peptidase (aminopeptidase and dipeptidase) activity increases with the advancing age of broilers.

Another possible factor contributing to the increasing apparent AA digestibility with age may be the greater absorptive area [39]. According to Nitsan et al. [33], the weight of the small intestine increased 10-fold at day 8 and 20-fold at day 23. Insoluble NSP is known to promote the development of gizzard [49]. As barley contains high insoluble NSP (122–142 g/kg) [6,27], it is likely that a more developed gizzard in barley will facilitate better nutrient digestion by mechanical breakdown of digesta. Well-developed GIT in older birds can overcome the negative viscosity effects of β-glucans [47].

Digesta retention time in the GIT plays a vital role in nutrient digestion because it determines the actual contact time among nutrients, digestive enzymes and microbiota. An increase in digesta retention enhances the absorption of nutrients by increasing the exposure time between the digesta and intestinal absorptive surface. Feed passage time through the GIT is reduced with bird age [50]. Increased digesta retention, in combination with a well-developed digestive tract, especially foregut, in older birds may also increase protein and AA digestibility by enhancing intestinal refluxes, subsequently re-exposing the digesta to pepsin [51].

Higher AA digestibility in barley with advancing age might be due to better tolerance of older birds to high NSP and viscosity owing to the increased stability of intestinal microbiota with advancing age [7]. Gut microbiota, a major consumer of AAs, may potentially influence digestion by competing for nutrients [52]. The GIT of the newly hatched chick is sterile. The primary source of initial microbiota is the farm environment where the hatchlings are reared. With increasing age, a cascade of changes occurs in microbiota including species diversity, followed by the complexity of population structure and finally maturation and stabilization. This process continues with age in commercial broilers, and the microbiota profile is stabilized by week 3 [53]. According to Choct [54], dietary NSP can promote microbial growth and fermentation. Increased degradation of soluble NSP and the resultant reduction in the viscosity with age may limit bacterial growth and partly mitigate the adverse effects caused by the excessive population of microbiota.

In contrast to the present findings, earlier age-related studies measuring the excreta or total tract AA digestibility in poultry [3,15,55] indicated higher protein and AA digestibility in younger birds. Fonolla et al. [3] reported a decline in the excreta digestibility of protein in broilers with advancing age (day 21 vs. 52), which was attributed to an increased excretion of metabolic N. A drop in true digestibilities of protein and AAs from 3 to 6 weeks of age was reported by Zuprizal et al. [15]. According to Carré et al. [55], the apparent protein digestibility in pea was lower in adult roosters than in 3-week-old broilers. Following feeding of a mixed diet with a wide variety of ingredients, ten Doeschate et al. [4] observed a reduction in protein digestibility coefficients with broiler age. The values were reported to be 0.849 (days 13 to 15), 0.830 (days 27 to 29) and 0.840 (days 41 to 43), respectively. However, these studies are based on total tract digestibility and are not comparable with current findings due to possible contamination of AAs from urine, microbial fermentation in the hindgut and, consequently, modifications in the AA constituents [5].

With advancing broiler age, the reduction in ileal digestibility of AAs along with other major nutrients in feed ingredients has also been reported. Adedokun et al. [18] reported 18.5% higher AIDC for TAAs in meat and bone meal in 5-day-old broilers (0.705) compared to that of 21-day broilers (0.595). This pattern remained unchanged even after the standardization of AIDC, with SIDC values of 0.760 and 0.632 at days 5 and 21, respectively. In a previous experiment [14], though there was no age influence on the average AIDC of TAAs in sorghum, the average SIDC of TAAs on day 7 (0.903) was 6.9% higher than on day 14 (0.844) and 8.5% higher than the average from day 21 to day 42 (0.819–0.854). Comparisons among published age-related SID digestibility estimates are not straightforward because of differences in ingredient type [1,13], assumed age-appropriate endogenous losses [21], secretion and activities of digestive enzymes [33], development and activity of gastrointestinal microbiota [53], environment, chemical nature of nutrients and methodology.

In the SIDC calculations, the AIDCs were corrected for the basal endogenous AA losses from various digestive, pancreatic and enzymatic secretions [56], and unsurprisingly, these two values were different with SIDC being higher. In practical feed formulations, the SIDC AA of ingredients is preferred because it is more additive especially when mixed into a complete diet. Moreover, standardization eliminates the underestimation of some nutritionally critical and less additive AAs in poultry diets [2]. Though there are some constraints in AIDC estimates, the AIDC is reported in this study to better understand the magnitude of age impact on the SIDC due to correction for age-dependent EAA losses. The differences between the AIDC and SIDC illustrated in Figure 1 and Figure 2 were due to the correction of apparent values by age-appropriate EAA losses determined in a previous study in our laboratory [21]. Although data [1,9,16] exist on the SID AA of different poultry feed ingredients, only a few studies [14,15,17,18,19,20] have determined the SID AA at different broiler ages. Except for some [14,18,19,20], all previous studies have used a single EAA flow value, derived from older birds, to standardize the AIDC values at different ages. From our previous findings [21], the basal EAA losses of TAAs at day 7 (12.93 g/kg DMI) were twice that of the average of days 14 to 35 (6.61 g/kg DMI) and almost three times higher than day 42 (4.48 g/kg DMI). These resulted in substantial differences in the percentage of increase in SIDC over AIDC (Table 11) after correction of apparent digestibility data with age-appropriate EAA losses. The current findings suggest that correcting AIDC AA using a single EAA flow value across ages underestimates the SIDC AA of feed ingredients in young birds and overestimates in older birds.

In general, the magnitude of increase in the SIDC AA of corn and barley after correction for age-appropriate EAAs decreased with advancing broiler age. After correcting for age-appropriate EAAs, an increase of 32.5% in the average SIDC of TAAs at day 7 was observed in corn that was almost two times higher compared to that from day 14 to 35 (13.9–17.0%) and three times higher than the increase at day 42 (9.85%). Similar to corn, the average SIDC of TAAs in barley at day 7 was 31.5% higher than the average AIDC of TAAs, which decreased to 7.17% at day 42.

To summarize, several factors such as the composition of grains, antinutritive factors, digestive tract development, secretion of enzymes, feed intake, nutrient load, digesta retention time and gut microbiota contribute to the age effect on the AA digestibility in broilers. The endogenous AA losses are a key factor with the greatest impact on the SIDC being observed in young broilers.

## 5. Conclusions

The present study provides information on the SIDC of AAs in corn and barley from hatching to the end of the broiler growth cycle. The findings suggest that the age influence on AA digestibility is dependent on the grain type and AA. The AIDC AA in corn increased with advancing age. The SIDC AA was higher at day 7, decreased at day 14 and increased and plateaued between days 21 and 35. A further decrease was observed at day 42. In the case of barley, both the AIDC and SIDC of AAs increased as the birds grew older. Standardization of AIDC AA with age-appropriate EAA flows resulted in marked differences in the SIDC of both grains. Application of a single EAA value for correction of the AIDC for broilers of different ages can result in the underestimation of the SIDC AA in young birds and overestimation in older birds. It is concluded that the precision of feed formulations can be improved by using age-specific EAA values for the standardization of AIDC AA values.

## Figures and Tables

**Figure 1 animals-11-03575-f001:**
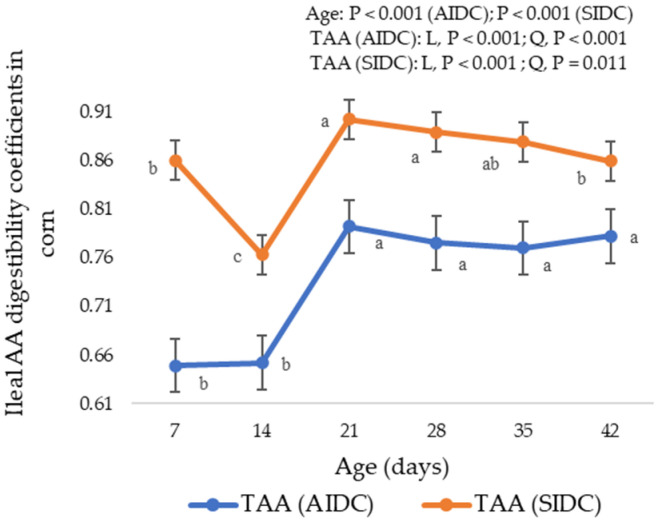
Apparent and standardized ileal digestibility coefficients of total amino acids (TAAs) in corn (bars represent means ± SE) as influenced by broiler age. ^a, b, c^ Values with different superscripts differ significantly (*p* < 0.05). L, Linear; Q, Quadratic.

**Figure 2 animals-11-03575-f002:**
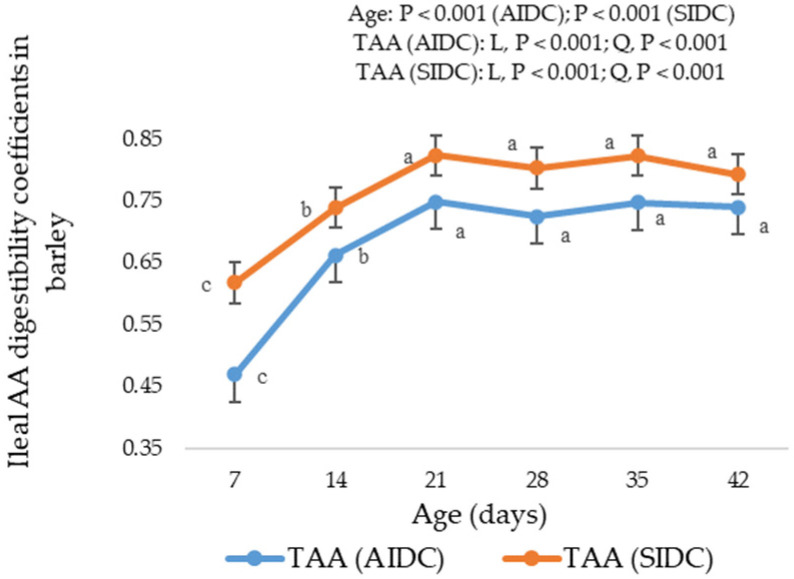
Apparent and standardized ileal digestibility coefficients of total amino acids in barley (bars represent means ± SE) as influenced by broiler age. ^a, b, c^ Values with different superscripts differ significantly (*p* < 0.05). L, Linear; Q, Quadratic.

**Table 1 animals-11-03575-t001:** Composition of the experimental diets (g/kg, as-fed basis).

Ingredient	Corn	Barley
Corn	938	-
Barley	-	938
Soybean oil	20	20
Dicalcium phosphate	18	18
Limestone	13	13
Titanium dioxide ^1^	5.0	5.0
Sodium chloride	2.0	2.0
Sodium bicarbonate	2.0	2.0
Trace mineral premix ^2^	1.0	1.0
Vitamin premix ^2^	1.0	1.0

^1^ Merck KGaA, Darmstadt, Germany. ^2^ Supplied per kilogram of diet: antioxidant (ethoxyquin), 100 mg; biotin, 0.2 mg; calcium pantothenate, 12.8 mg; cholecalciferol, 0.06 mg; cyanocobalamin, 0.017 mg; folic acid, 5.2 mg; menadione, 4 mg; niacin, 35 mg; pyridoxine, 10 mg; trans-retinol, 3.33 mg; riboflavin, 12 mg; thiamine, 3.0 mg; dl-α-tocopheryl acetate, 60 mg; choline chloride, 638 mg; Co, 0.3 mg; Cu, 3.0 mg; Fe, 25 mg; I, 1 mg; Mn, 125 mg; Mo, 0.5 mg; Se, 0.2 mg; Zn, 60 mg.

**Table 2 animals-11-03575-t002:** Composition and calculated analysis (g/kg, as-fed basis) of broiler starter and finisher diets.

Ingredient	Starter Diet (0–21 days)	Finisher Diet (22–42 days)
Corn	574.2	660
Soybean meal, 460 g/kg	381.4	295.6
Soybean oil	8.8	13.6
Limestone	11.3	9.9
Dicalcium phosphate	10.7	8.2
DL-methionine	3.3	3.0
L-lysine HCl	2.0	1.9
L-threonine	1.0	0.7
Sodium bicarbonate	2.7	2.5
Sodium chloride	2.5	2.5
Trace mineral premix ^1^	1.0	1.0
Vitamin premix ^1^	1.0	1.0
Phytase	0.1	0.1
Calculated analysis		
Apparent metabolizable energy (MJ/kg)	12.14	12.69
Crude protein	225	190
Digestible lysine	11.0	9.2
Digestible methionine	6.2	5.6
Digestible methionine + cysteine	9.2	8.3
Digestible threonine	7.2	6.0
Crude fat	32	39
Crude fiber	29.3	27.5
Calcium	9.8	8.5
Available phosphorus	4.9	4.2
Sodium	2.2	2.1
Chloride	2.3	2.3
Potassium	11.5	9.7

^1^ Supplied per kilogram of diet: antioxidant (ethoxyquin), 100 mg; biotin, 0.2 mg; calcium pantothenate, 12.8 mg; cholecalciferol, 0.06 mg; cyanocobalamin, 0.017 mg; folic acid, 5.2 mg; menadione, 4 mg; niacin, 35 mg; pyridoxine, 10 mg; trans-retinol, 3.33 mg; riboflavin, 12 mg; thiamine, 3.0 mg; dl-α-tocopheryl acetate, 60 mg; choline chloride, 638 mg; Co, 0.3 mg; Cu, 3.0 mg; Fe, 25 mg; I, 1 mg; Mn, 125 mg; Mo, 0.5 mg; Se, 0.2 mg; Zn, 60 mg.

**Table 3 animals-11-03575-t003:** Proximate, carbohydrate and amino acid composition of grains (g/kg, as-received basis).

Item	Corn	Barley
Dry matter	859	875
Starch	590	541
Nitrogen (N)	10.8	18.4
Crude protein (N × 6.25)	67.8	115
Fat	32.4	21.0
Neutral detergent fiber	83.1	110
Gross energy (MJ/kg)	16.3	16.3
Ash	20.5	18.6
Calcium	0.17	0.14
Phosphorus	2.47	3.02
Indispensable amino acids (IAAs)		
Arg	3.43	5.28
His	2.07	2.37
Ile	2.49	3.98
Leu	8.14	7.53
Lys	2.33	3.72
Met	1.31	1.77
Thr	2.59	3.54
Trp	0.58	1.35
Val	3.44	5.52
Total IAA	26.4	35.1
Dispensable amino acids (DAAs)		
Ala	5.14	4.54
Asp	4.77	6.78
Cys ^1^	1.37	2.23
Glu	12.4	24.3
Gly ^1^	2.85	4.44
Pro	5.60	10.8
Ser	3.29	4.41
Total DAA	35.4	57.5
Total AA ^2^	61.8	92.5

^1^ Semi-indispensable amino acids for poultry. ^2^ Total AA = IAA + DAA.

**Table 4 animals-11-03575-t004:** Daily feed intake (DFI; g/bird/d), daily weight gain (DWG; g/bird/d), gizzard pH and viscosity (cP) in jejunal digesta of broilers fed corn- and barley-based diets at different ages ^1^.

	Age (Days)							Orthogonal Polynomial Contrasts	
Parameter	7	14	21	28	35	42	Pooled SEM	Linear	Quadratic
Corn									
DFI ^2^	12.1	34.8	77.9	123	135	165	1.249	0.001	0.001
DWG ^2^	10.5	28.4	35.8	54.3	58.9	59.7	0.476	0.001	0.001
Gizzard pH ^3^	2.53	2.07	2.24	2.59	3.19	3.61	0.108	0.001	0.001
Viscosity ^3^	2.03	2.33	2.23	2.25	2.11	2.34	0.079	0.168	0.480
Barley									
DFI ^2^	13.6	37.3	84.8	129	142	170	1.153	0.001	0.001
DWG ^2^	12.5	31.4	38.8	56.2	61.1	62.9	0.855	0.001	0.001
Gizzard pH ^3^	2.14	2.08	2.75	3.24	3.38	3.05	0.105	0.001	0.001
Viscosity ^3^	2.94	2.70	2.69	2.80	2.75	2.94	0.093	0.745	0.024

^1^ Each value represents the mean of six replicates (14, 12 and 10 birds per replicate for 7-, 14- and 21-days old birds, respectively; eight birds per replicate for 28- and 35-days old birds; and six birds per replicate for 42-days old birds). ^2^ Measured during 4-day feeding of experimental diets. ^3^ Calculated as the mean of six replicates (two birds per replicate).

**Table 5 animals-11-03575-t005:** Apparent ileal digestibility coefficients ^1^ of nitrogen (N) and amino acids of corn at different ages of broilers ^1^.

	Age (Days)							Orthogonal Polynomial Contrasts	
Parameter	7	14	21	28	35	42	Pooled SEM	Linear	Quadratic
N	0.651	0.669	0.779	0.766	0.764	0.766	0.0089	0.001	0.001
Indispensable amino acids									
Arg	0.745	0.752	0.864	0.849	0.850	0.859	0.0056	0.001	0.001
His	0.718	0.744	0.805	0.795	0.792	0.802	0.0080	0.001	0.001
Ile	0.628	0.645	0.805	0.779	0.771	0.789	0.0100	0.001	0.001
Leu	0.799	0.817	0.901	0.882	0.872	0.882	0.0059	0.001	0.001
Lys	0.467	0.473	0.753	0.709	0.701	0.712	0.0142	0.001	0.001
Met	0.724	0.721	0.882	0.856	0.850	0.859	0.0098	0.001	0.001
Thr	0.464	0.435	0.636	0.635	0.635	0.648	0.0138	0.001	0.001
Trp	0.464	0.469	0.648	0.635	0.654	0.662	0.0143	0.001	0.001
Val	0.604	0.616	0.794	0.779	0.768	0.786	0.0100	0.001	0.001
IAA	0.624	0.630	0.788	0.769	0.766	0.778	0.0093	0.001	0.001
Dispensable amino acids									
Ala	0.771	0.779	0.874	0.854	0.839	0.846	0.0072	0.001	0.001
Asp	0.609	0.601	0.773	0.753	0.739	0.753	0.0100	0.001	0.001
Cys ^2^	0.709	0.737	0.747	0.738	0.730	0.747	0.0127	0.077	0.209
Glu	0.794	0.798	0.893	0.869	0.862	0.866	0.0067	0.001	0.001
Gly ^2^	0.551	0.534	0.724	0.708	0.703	0.721	0.0121	0.001	0.001
Pro	0.706	0.725	0.811	0.801	0.798	0.813	0.0079	0.001	0.001
Ser	0.625	0.589	0.769	0.757	0.761	0.772	0.0119	0.001	0.001
DAA	0.681	0.680	0.799	0.783	0.776	0.788	0.0089	0.001	0.001
TAA	0.649	0.652	0.792	0.775	0.770	0.782	0.0087	0.001	0.001

^1^ Each value represents the mean of six replicates (14, 12 and 10 birds per replicate for 7-, 14- and 21-days old birds, respectively; eight birds per replicate for 28- and 35-days old birds; and six birds per replicate for 42-days old birds). ^2^ Semi-indispensable amino acids for poultry. DAA = Average digestibility of dispensable amino acids; IAA = Average digestibility of indispensable amino acids; TAA = Average digestibility of all amino acids.

**Table 6 animals-11-03575-t006:** Standardized ileal digestibility coefficients ^1^ of nitrogen (N) and amino acids of corn at different ages of broilers ^2^.

	Age (Days)							Orthogonal Polynomial Contrasts	
Parameter	7	14	21	28	35	42	Pooled SEM	Linear	Quadratic
N	0.936	0.817	0.922	0.911	0.907	0.868	0.0089	0.295	0.684
Indispensable amino acids									
Arg	0.914	0.828	0.943	0.939	0.930	0.909	0.0056	0.001	0.003
His	0.839	0.810	0.866	0.862	0.854	0.844	0.0080	0.034	0.039
Ile	0.844	0.757	0.917	0.898	0.875	0.859	0.0100	0.001	0.001
Leu	0.902	0.868	0.952	0.939	0.923	0.915	0.0059	0.001	0.001
Lys	0.704	0.583	0.858	0.827	0.805	0.776	0.0142	0.001	0.001
Met	0.899	0.801	0.963	0.948	0.927	0.904	0.0097	0.001	0.001
Thr	0.912	0.678	0.872	0.856	0.871	0.821	0.0138	0.348	0.119
Trp	0.769	0.640	0.828	0.823	0.834	0.792	0.0143	0.001	0.048
Val	0.807	0.723	0.902	0.892	0.873	0.861	0.0100	0.001	0.001
IAA	0.843	0.743	0.900	0.887	0.877	0.854	0.0091	0.001	0.002
Dispensable amino acids									
Ala	0.896	0.841	0.934	0.923	0.901	0.887	0.0072	0.043	0.001
Asp	0.863	0.733	0.902	0.887	0.868	0.843	0.0100	0.002	0.017
Cys ^3^	0.968	0.906	0.908	0.894	0.895	0.877	0.0089	0.001	0.013
Glu	0.912	0.853	0.948	0.931	0.918	0.903	0.0067	0.026	0.002
Gly ^3^	0.784	0.654	0.845	0.836	0.823	0.807	0.0121	0.001	0.034
Pro	0.845	0.799	0.885	0.874	0.871	0.866	0.0079	0.001	0.048
Ser	0.902	0.733	0.915	0.897	0.907	0.876	0.0119	0.001	0.996
DAA	0.881	0.788	0.905	0.892	0.883	0.865	0.0083	0.010	0.109
TAA	0.860	0.763	0.902	0.889	0.879	0.859	0.0087	0.001	0.011

^1^ Apparent digestibility values were standardized using the following basal ileal endogenous flow values (g/kg DM intake), determined by feeding nitrogen-free diet at different ages [21]: Day 7: N, 3.59; Arg, 0.68; His, 0.29; Ile, 0.63; Leu, 0.97; Lys, 0.64; Met, 0.27; Thr, 1.35; Trp, 0.21; Val, 0.81; Ala, 0.75; Asp, 1.41; Cys, 0.47; Glu, 1.71; Gly, 0.78; Pro, 0.91; and Ser, 1.06. Day 14: N, 1.87; Arg, 0.30; His, 0.16; Ile, 0.33; Leu, 0.49; Lys, 0.30; Met, 0.12; Thr, 0.73; Trp, 0.12; Val, 0.43; Ala, 0.37; Asp, 0.73; Cys, 0.27; Glu, 0.80; Gly, 0.39; Pro, 0.48; and Ser, 0.55. Day 21: N, 1.79; Arg, 0.31; His, 0.15; Ile, 0.32; Leu, 0.49; Lys, 0.28; Met, 0.12; Thr, 0.71; Trp, 0.12; Val, 0.43; Ala, 0.36; Asp, 0.72; Cys, 0.26; Glu, 0.80; Gly, 0.40; Pro, 0.48; and Ser, 0.56. Day 28: N, 1.82; Arg, 0.36; His, 0.16; Ile, 0.34; Leu, 0.55; Lys, 0.32; Met, 0.14; Thr, 0.67; Trp, 0.13; Val, 0.45; Ala, 0.41; Asp, 0.74; Cys, 0.25; Glu, 0.89; Gly, 0.43; Pro, 0.48; and Ser, 0.54. Day 35: N, 1.81; Arg, 0.32; His, 0.15; Ile, 0.30; Leu, 0.49; Lys, 0.28; Met, 0.12; Thr, 0.71; Trp, 0.12; Val, 0.42; Ala, 0.37; Asp, 0.72; Cys, 0.26; Glu, 0.81; Gly, 0.40; Pro, 0.47; and Ser, 0.56. Day 42: N, 1.29; Arg, 0.20; His, 0.10; Ile, 0.20; Leu, 0.31; Lys, 0.18; Met, 0.07; Thr, 0.52; Trp, 0.09; Val, 0.29; Ala, 0.25; Asp, 0.49; Cys, 0.21; Glu, 0.53; Gly, 0.28; Pro, 0.34; and Ser, 0.40. ^2^ Each value represents the mean of six replicates (14, 12 and 10 birds per replicate for 7-, 14- and 21-days old birds, respectively; eight birds per replicate for 28- and 35-days old birds; and six birds per replicate for 42-days old birds). ^3^ Semi-indispensable amino acids for poultry. DAA = Average digestibility of dispensable amino acids; IAA = Average digestibility of indispensable amino acids; TAA = Average digestibility of all amino acids.

**Table 7 animals-11-03575-t007:** Influence of age (days) on standardized digestible protein (CP) and amino acid contents ^1^ (g/kg) of corn, as-received basis.

	Age (Days)							Orthogonal Polynomial Contrasts	
Parameter	7	14	21	28	35	42	Pooled SEM	Linear	Quadratic
CP	63.4	55.4	62.5	61.7	61.5	58.9	0.607	0.295	0.685
Indispensable amino acids									
Arg	3.14	2.84	3.23	3.22	3.19	3.12	0.019	0.001	0.003
His	1.74	1.68	1.79	1.78	1.77	1.75	0.017	0.034	0.039
Ile	2.10	1.88	2.28	2.24	2.18	2.14	0.025	0.001	0.001
Leu	7.34	7.06	7.75	7.65	7.51	7.45	0.048	0.001	0.001
Lys	1.64	1.36	1.99	1.93	1.88	1.81	0.033	0.001	0.001
Met	1.18	1.05	1.26	1.24	1.21	1.18	0.013	0.001	0.001
Thr	2.36	1.75	2.26	2.22	2.26	2.13	0.036	0.348	0.119
Trp	0.45	0.37	0.48	0.48	0.48	0.46	0.008	0.001	0.048
Val	2.78	2.49	3.10	3.07	3.00	2.96	0.034	0.001	0.001
IAA	22.7	20.5	24.2	23.8	23.5	22.9	0.22	0.001	0.001
Dispensable amino acids									
Ala	4.60	4.32	4.80	4.74	4.63	4.56	0.037	0.043	0.001
Asp	4.11	3.49	4.30	4.23	4.14	4.02	0.048	0.002	0.017
Cys ^2^	1.33	1.24	1.24	1.22	1.23	1.20	0.012	0.001	0.013
Glu	11.3	10.6	11.8	11.5	11.4	11.2	0.083	0.026	0.002
Gly ^2^	2.24	1.86	2.41	2.38	2.35	2.29	0.034	0.001	0.034
Pro	4.73	4.47	4.96	4.89	4.88	4.85	0.044	0.001	0.047
Ser	2.97	2.41	3.01	2.95	2.99	2.88	0.039	0.001	0.996
DAA	31.2	28.4	32.5	31.9	31.6	31.0	0.28	0.002	0.018
TAA	54.1	48.9	56.6	55.8	55.1	53.9	0.49	0.001	0.005

^1^ Each value represents the mean of six replicates (14, 12 and 10 birds per replicate for 7-, 14- and 21-days old birds, respectively; eight birds per replicate for 28- and 35-days old birds; and six birds per replicate for 42-days old birds). ^2^ Semi-indispensable amino acids for poultry. DAA = Average digestibility of dispensable amino acids; IAA = Average digestibility of indispensable amino acids; TAA = Average digestibility of all amino acids.

**Table 8 animals-11-03575-t008:** Apparent ileal digestibility coefficients ^1^ of nitrogen (N) and amino acids of barley at different ages of broilers ^1^.

	Age (Days)							Orthogonal Polynomial Contrasts	
Parameter	7	14	21	28	35	42	Pooled SEM	Linear	Quadratic
N	0.504	0.670	0.731	0.713	0.730	0.720	0.0142	0.001	0.001
Indispensable amino acids									
Arg	0.611	0.705	0.803	0.772	0.802	0.791	0.0135	0.001	0.001
His	0.544	0.703	0.726	0.702	0.729	0.719	0.0142	0.001	0.001
Ile	0.436	0.660	0.749	0.725	0.747	0.749	0.0155	0.001	0.001
Leu	0.537	0.717	0.794	0.766	0.788	0.783	0.0133	0.001	0.001
Lys	0.230	0.545	0.739	0.682	0.717	0.717	0.0211	0.001	0.001
Met	0.479	0.692	0.812	0.777	0.796	0.789	0.0186	0.001	0.001
Thr	0.261	0.524	0.646	0.633	0.669	0.655	0.0196	0.001	0.001
Trp	0.430	0.646	0.709	0.694	0.724	0.713	0.0163	0.001	0.001
Val	0.449	0.652	0.756	0.737	0.753	0.752	0.0144	0.001	0.001
IAA	0.442	0.649	0.748	0.721	0.747	0.741	0.0155	0.001	0.001
Dispensable amino acids									
Ala	0.423	0.633	0.734	0.699	0.722	0.717	0.0158	0.001	0.001
Asp	0.328	0.569	0.703	0.680	0.707	0.699	0.0179	0.001	0.001
Cys ^2^	0.620	0.764	0.743	0.731	0.739	0.728	0.0171	0.004	0.001
Glu	0.709	0.812	0.845	0.822	0.832	0.825	0.0107	0.001	0.001
Gly ^2^	0.360	0.561	0.675	0.658	0.676	0.673	0.0158	0.001	0.001
Pro	0.681	0.782	0.805	0.793	0.803	0.799	0.0112	0.001	0.001
Ser	0.413	0.622	0.727	0.708	0.741	0.728	0.0159	0.001	0.001
DAA	0.505	0.678	0.747	0.727	0.746	0.738	0.0140	0.001	0.001
TAA	0.469	0.662	0.748	0.724	0.747	0.739	0.0148	0.001	0.001

^1^ Each value represents the mean of six replicates (14, 12 and 10 birds per replicate for 7-, 14- and 21-days old birds, respectively; eight birds per replicate for 28- and 35-days old birds; and six birds per replicate for 42-days old birds). ^2^ Semi-indispensable amino acids for poultry. DAA = Average digestibility of dispensable amino acids; IAA = Average digestibility of indispensable amino acids; TAA = Average digestibility of all amino acids.

**Table 9 animals-11-03575-t009:** Standardized ileal digestibility coefficients ^1^ of nitrogen (N) and amino acids of barley at different ages of broilers ^2^.

	Age (Days)							Orthogonal Polynomial Contrasts	
Parameter	7	14	21	28	35	42	Pooled SEM	Linear	Quadratic
N	0.674	0.759	0.816	0.799	0.816	0.782	0.0143	0.001	0.001
Indispensable amino acids									
Arg	0.724	0.755	0.855	0.831	0.854	0.824	0.0135	0.001	0.001
His	0.652	0.762	0.780	0.762	0.783	0.756	0.0143	0.001	0.001
Ile	0.574	0.732	0.821	0.801	0.813	0.793	0.0155	0.001	0.001
Leu	0.650	0.773	0.851	0.829	0.844	0.819	0.0133	0.001	0.001
Lys	0.382	0.616	0.806	0.757	0.783	0.758	0.0211	0.001	0.001
Met	0.611	0.752	0.874	0.847	0.854	0.823	0.0186	0.001	0.001
Thr	0.595	0.705	0.822	0.798	0.845	0.784	0.0196	0.001	0.001
Trp	0.564	0.721	0.787	0.776	0.802	0.769	0.0163	0.001	0.001
Val	0.578	0.719	0.825	0.809	0.819	0.799	0.0144	0.001	0.001
IAA	0.592	0.726	0.825	0.801	0.822	0.792	0.0155	0.001	0.001
Dispensable amino acids									
Ala	0.566	0.703	0.803	0.778	0.793	0.764	0.0158	0.001	0.001
Asp	0.509	0.664	0.796	0.776	0.800	0.763	0.0179	0.001	0.001
Cys ^3^	0.806	0.870	0.844	0.829	0.843	0.809	0.0171	0.553	0.045
Glu	0.771	0.841	0.874	0.854	0.861	0.844	0.0107	0.001	0.001
Gly ^3^	0.513	0.639	0.753	0.742	0.755	0.729	0.0158	0.001	0.001
Pro	0.755	0.821	0.844	0.831	0.842	0.827	0.0112	0.001	0.001
Ser	0.623	0.732	0.838	0.815	0.852	0.808	0.0159	0.001	0.001
DAA	0.649	0.753	0.822	0.804	0.821	0.792	0.0140	0.001	0.001
TAA	0.617	0.738	0.823	0.802	0.822	0.792	0.0148	0.001	0.001

^1^ Apparent digestibility values were standardized using the following basal ileal endogenous flow values (g/kg DM intake), determined by feeding nitrogen-free diet at different ages [21]; see Table 6. ^2^ Each value represents the mean of six replicates (14, 12 and 10 birds per replicate for 7-, 14- and 21-days old birds, respectively; eight birds per replicate for 28- and 35-days old birds; and six birds per replicate for 42-days old birds). ^3^ Semi-indispensable amino acids for poultry. DAA = Average digestibility of dispensable amino acids; IAA = Average digestibility of indispensable amino acids; TAA = Average digestibility of all amino acids.

**Table 10 animals-11-03575-t010:** Influence of age (days) on standardized digestible protein (CP) and amino acid contents ^1^ (g/kg) of barley, as-received basis.

	Age (Days)							Orthogonal Polynomial Contrasts	
Parameter	7	14	21	28	35	42	Pooled SEM	Linear	Quadratic
CP	77.7	87.5	94.1	92.1	92.9	90.1	1.57	0.001	0.001
Indispensable amino acids									
Arg	3.82	3.99	4.51	4.39	4.51	4.35	0.071	0.001	0.001
His	1.55	1.81	1.85	1.81	1.86	1.79	0.034	0.001	0.001
Ile	2.28	2.91	3.27	3.19	3.24	3.16	0.062	0.001	0.001
Leu	4.89	5.82	6.41	6.25	6.36	6.17	0.100	0.001	0.001
Lys	1.42	2.29	2.99	2.82	2.91	2.82	0.079	0.001	0.001
Met	1.08	1.33	1.55	1.49	1.51	1.46	0.033	0.001	0.001
Thr	2.10	2.49	2.91	2.82	2.99	2.78	0.069	0.001	0.001
Trp	0.76	0.97	1.06	1.05	1.08	1.04	0.021	0.001	0.001
Val	3.19	3.97	4.55	4.46	4.53	4.41	0.079	0.001	0.001
IAA	21.1	25.6	29.1	28.3	28.9	27.9	0.53	0.001	0.001
Dispensable amino acids									
Ala	2.57	3.19	3.65	3.53	3.60	3.47	0.072	0.001	0.001
Asp	3.45	4.50	5.39	5.26	5.43	5.17	0.122	0.001	0.001
Cys ^2^	1.79	1.94	1.88	1.85	1.88	1.80	0.038	0.553	0.044
Glu	18.8	20.5	21.2	20.8	20.9	20.5	0.259	0.001	0.001
Gly ^2^	2.28	2.84	3.34	3.29	3.35	3.24	0.070	0.001	0.001
Pro	8.12	8.84	9.08	8.94	9.06	8.89	0.121	0.001	0.001
Ser	2.75	3.23	3.69	3.59	3.76	3.56	0.069	0.001	0.001
DAA	38.1	43.3	46.6	45.6	48.0	45.0	0.70	0.001	0.001
TAA	60.8	70.6	77.4	75.5	76.9	74.6	1.25	0.001	0.001

^1^ Each value represents the mean of six replicates (14, 12 and 10 birds per replicate for 7-, 14- and 21-days old birds, respectively; eight birds per replicate for 28- and 35-days old birds; and six birds per replicate for 42-days old birds). ^2^ Semi-indispensable amino acids for poultry. DAA = Average digestibility of dispensable amino acids; IAA = Average digestibility of indispensable amino acids; TAA = Average digestibility of all amino acids.

**Table 11 animals-11-03575-t011:** Percentage increase in digestibility coefficients of nitrogen (N) and amino acids in corn and barley after correction of apparent ileal digestibility coefficients for age-appropriate endogenous amino acid losses of broilers.

	Corn						Barley					
	Age (Days)						Age (Days)					
Parameter	7	14	21	28	35	42	7	14	21	28	35	42
N	43.8	22.1	18.4	18.9	18.7	13.3	33.7	13.3	11.6	12.1	11.8	8.61
Indispensable amino acids												
Arg	22.7	10.1	9.14	10.6	9.41	5.82	18.5	7.09	6.48	7.64	6.48	4.17
His	16.9	8.87	7.58	8.43	7.83	5.24	19.9	8.39	7.44	8.55	7.41	5.15
Ile	34.4	17.4	13.9	15.3	13.5	8.87	31.7	10.9	9.61	10.5	8.84	5.87
Leu	12.9	6.24	5.66	6.46	5.85	3.74	21.0	7.81	7.18	8.22	7.11	4.60
Lys	50.7	23.3	13.9	16.6	14.8	8.99	66.1	13.0	9.07	11.0	9.21	5.72
Met	24.2	11.1	9.18	10.8	9.06	5.24	27.6	8.67	7.64	9.01	7.29	4.31
Thr	96.6	55.9	37.1	34.8	37.2	26.7	128	34.5	27.2	26.1	26.3	19.7
Trp	65.7	36.5	27.8	29.6	27.5	19.6	31.2	11.6	11.0	11.8	10.8	7.85
Val	33.6	17.4	13.6	14.5	13.7	9.54	28.7	10.3	9.13	9.77	8.76	6.25
IAA	35.1	17.9	14.2	14.0	14.5	9.77	33.9	11.9	10.3	11.1	10.0	6.88
Dispensable amino acids												
Ala	16.2	7.96	6.90	8.08	7.39	4.85	33.8	11.1	9.40	11.3	9.83	6.56
Asp	41.7	21.9	16.7	17.8	17.5	11.9	55.2	16.7	13.2	14.1	13.2	9.16
Cys ^1^	36.5	22.9	21.6	21.1	22.6	17.4	30.0	13.9	13.6	13.4	14.1	11.1
Glu	14.9	6.89	6.16	7.13	6.50	4.27	8.74	3.57	3.43	3.89	3.49	2.30
Gly ^1^	42.3	22.5	16.7	18.1	17.1	11.9	42.5	13.9	11.6	12.8	11.7	8.32
Pro	19.7	10.2	9.12	9.11	9.15	6.52	10.9	4.99	4.84	4.79	4.86	3.50
Ser	44.3	24.5	18.9	18.5	19.2	13.5	50.8	17.7	15.3	15.1	14.9	10.9
DAA	29.4	15.9	13.3	13.9	13.8	9.77	28.5	11.1	10.0	10.6	10.1	7.32
TAA	32.5	17.0	13.9	14.7	14.2	9.85	31.6	11.5	10.0	10.8	10.0	7.17

^1^ Semi-indispensable amino acids for poultry. DAA = Average digestibility of dispensable amino acids; IAA = Average digestibility of indispensable amino acids; TAA = Average digestibility of all amino acids.

## Data Availability

All available data are incorporated in the manuscript.

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
