# Peer review of "Influence of Age on the Standardized Ileal Amino Acid Digestibility of Corn and Barley in Broilers"

_animals, 2021, doi:10.3390/ani11123575_

Round 1
Reviewer 1 Report
The objective of the work was to determine standardized ileal digestibility of amino acids in corn and barley in broilers at various ages. The paper is well-written and worthy of publication. There are however a few things the authors should address.
There are some grammatical errors in the Simple Summary. The authors should review this section.
Line 24 The statement that SIDC was higher at day 7 and lower at day 14 for corn is a bit troubling. Since Apparent digestibility did not show this, I wonder if it is a real effect. I suggest not emphasizing this.
Line 102-106 I suggest that Table 3 should come before Table 2 as it is mentioned in the methods first.
Line 140-142 It is not clear what happened to the upper ileal digesta. Were the upper and lower samples analyzed separately?
Table 3 Do these values represent a single analysis of each grain or more than one? Please clarify.
Line 227 This needs to be clarified. My understanding (from line 119) was that all birds were fed the starter and finisher diets shown in Table 2 until they were switched to the experimental diets at each age for a 4 day study period. It is not clear why there are separate sections for corn and barley fed birds in Table 4. The starter and finisher diets were corn based and so why show barley?
Table 6 and 9 It is difficult to understand how these effects have a significant linear relationship. For example, the values on day 7, 14, 21, 28,35 and 42 for lysine in corn are:0.70, 0.58, 0.86, 0.83, 0.81 and 0.78. Similar results are seen for the total indispensable AA. While there is an unexplained dip in day 14 compared to days 7 and 21, there is really not much variation that I would call a linear relationship.
Author Response
"Please see the attachment"

Reviewer 2 Report
Overall the work is well presented and the results are of use to feed formulators
Some minor changes are suggested
Ln 16: remove ‘Number’ add ‘A range’
Ln 18: remove ‘get’ add ‘source’
Ln 239: ‘vis cosity (cP)’ remove space
Ln 378: remove ‘fed’ add ’when feeding’
Ln 392: ‘moderate’ a P value of P < 0.001) suggests better than moderate
Ln 396: remove ‘might’ add ‘that’
ln 400: ‘Yu et al. [35] reported the duodenal digesta viscosity in 3- and 6-weeks old broilers at different replacement levels of corn with barley, and observed elevated viscosity with increasing barley inclusions.’ This point needs greater clarity
Ln 415: remove ‘feeding’ add ‘fed’
Ln 417: remove ‘eight-weeks’ add ‘eight-weeks of age’
Ln 424: ‘supply organs’ what is meant by supply organs?
Ln 440: remove ‘contained wide’ add ‘containing a wide’
Ln 461: remove ‘in add ‘for’
Ln 551: remove ‘Following feeding mixed diet’ add ‘Following the feeding of a mixed diet’
Ln 562: remove ‘though’
Ln 576: remove ‘for better understanding´ add ‘to better understand’
Author Response
"Please see the attachment."
